# Enhance the Accuracy of Landslide Detection in UAV Images Using an Improved Mask R-CNN Model: A Case Study of Sanming, China

**DOI:** 10.3390/s23094287

**Published:** 2023-04-26

**Authors:** Lu Yun, Xinxin Zhang, Yuchao Zheng, Dahan Wang, Lizhong Hua

**Affiliations:** 1College of Computer and Information Engineering, Xiamen University of Technology, Xiamen 361024, China; 2Fujian Key Laboratory of Pattern Recognition and Image Understanding, Xiamen University of Technology, Xiamen 361024, China

**Keywords:** landslide, deep learning, CBAM, GA-RPN, Mask R-CNN

## Abstract

Extracting high-accuracy landslide areas using deep learning methods from high spatial resolution remote sensing images is a hot topic in current research. However, the existing deep learning algorithms are affected by background noise and landslide scale effects during the extraction process, leading to poor feature extraction effects. To address this issue, this paper proposes an improved mask regions-based convolutional neural network (Mask R-CNN) model to identify the landslide distribution in unmanned aerial vehicles (UAV) images. The improvement of the model mainly includes three aspects: (1) an attention mechanism of the convolutional block attention module (CBAM) is added to the backbone residual neural network (ResNet). (2) A bottom-up channel is added to the feature pyramidal network (FPN) module. (3) The region proposal network (RPN) is replaced by guided anchoring (GA-RPN). Sanming City, China was selected as the study area for the experiments. The experimental results show that the improved model has a recall of 91.4% and an accuracy of 92.6%, which is 12.9% and 10.9% higher than the original Mask R-CNN model, respectively, indicating that the improved model is more effective in landslide extraction.

## 1. Introduction

Due to their frequent occurrence, landslide geological hazards threaten the ecological environment and society worldwide. Landslides are particularly widespread in 70% of China’s regions, with numerous landslides in the mountainous areas of Southwestern, Northwestern, Eastern, Southern, and Central China, as well as in the hilly and Loess Plateau areas; thus, China is seriously affected by landslides [1]. Among all types of geological hazards, landslides rank first in terms of human casualties and economic losses and impact both local construction and long-term development [2].

Therefore, immediate access to information related to a landslide occurrence area is the key to mitigating the disaster impact attributable to the occurrence of landslides. In landslide hazard research, landslide identification and mapping are the basis for other research areas [3]. Traditional landslide identification mainly relies on ground survey data and requires experts with knowledge of landslide geology theory to visit the disaster site to investigate and determine the scale of the landslide area and the surrounding disaster situation, which is time-consuming and often incomplete, and some landslide areas may experience secondary landslides and other geological hazards with limited site investigation [4]. Hence, landslide identification is now mainly carried out using remote sensing images.

Advances in spatial information technology have facilitated the acquisition of high-resolution landslide imagery and the processing of landslide information. Currently, high-resolution imagery of landslide hazards is mainly derived from satellite remote sensing or from locating areas by UAV [5]. Satellite remote sensing observation information is macroscopic and comprehensive and can be continuously observed for a long time to form time series information at a low cost. Researchers initially used visual interpretation, but this required both specialists in the relevant disciplines and was inefficient [6]. Advances in machine learning have changed this; common machine learning identification algorithms include the maximum likelihood ratio (MLR), support vector machine (SVM) [7], multilayer perceptron neural network (MLP-NN) [8], recurrent neural network (RNN) [9], random forest (RF) [10], etc. These algorithms have proven to be very effective in landslide information extraction [11], for example, Zêzere [12], Huang [13], and other scholars performed a comparison of machine learning methods and concluded that SVM models are reliably effective. However, satellite data are often disturbed by clouds and fog, the timeliness is not enough, and the data accuracy of general satellites is insufficient. Since UAVs are light and flexible, and close to the ground, they have high resolution and are not disturbed by weather, such as clouds, and have good timeliness and high accuracy in terrain modeling [14]. Qi et al. [15] combined UAV remote sensing image technology and machine learning technology to construct a landslide feature extraction system and integrated it with experimental design to verify the performance of the system. While Ghorbanzadeh et al. [16] studies show that almost all machine learning methods used to analyze the potential risk of landslides rely heavily on historical datasets of the spatial extent of landslide areas, or feature datasets of GPS point locations for each landslide in at least one study area.

The development of deep learning as a branch of machine learning has produced a better solution to this problem. Convolutional neural networks (CNNs) have been utilized for the classification and segmentation of various images in computer vision [17,18], semantic segmentation [19], scene annotation using different high spatial resolutions and aerial images [20], and target detection analysis [21]. In neural-network-based landslide recognition algorithms, Bui et al. [22] applied bidimensional empirical mode decomposition (BEMD) to image preprocessing of landslides to determine the features of landslides and trained the convolutional neural network structure to correctly identify landslide features. Landslide identification using CNN or the self-developed residual neural unity network (Res-U-Net) are applications of deep learning in the field of landslide detection [23,24]. Yang et al. showed that the addition of CBAM to Res-U-Net can optimize the model, highlight important features, and handle complex landslide patterns, but how to effectively utilize and optimize the feature maps generated by CNN for the transformer and avoid the loss of global contextual information remains a subsequent problem to be addressed [25]. Cheng et al. [26] developed the you only look once-small attention (YOLO-SA) model with a small number of parameters for landslide image recognition based on the one-stage detector YOLOv4, which achieved the goal of high accuracy with few parameters. Adriana Romero et al. [27] applied deep convolutional networks to landslide recognition; the results suggested that deep structures are superior to single-level structures. Mask R-CNN is often used to achieve multiclass target detection in remote sensing images [28,29]. Sui et al. [30] proposed a method to fuse the Mask R-CNN model with an attention mechanism and image multiscale segmentation algorithm for post-earthquake building façade damage detection. Zhang et al. [31] employed YOLO, faster region-based convolutional neural network (Faster R-CNN) [32], and single shot multi-box detector (SSD) [33] to detect landslide images and discussed the characteristics of the three algorithms and the application of parameters in the detection process to obtain better detection results, while the shape of the landslide was not specifically identified. Liu et al. [34] added bottom-up channels to FPN and found that it was possible to leverage low-level localization and high-level semantic information, but mainly for post-earthquake landslide data. Xi et al. [35] considered the influence of landslide images and features with similar characteristics, such as bare ground, to simulate a difficult sample of landslides for detection, which reduced the landslide false detection rate. However, the processing of landslide data is more complicated. Fu et al. [36] used the method of transfer learning to build a recognition model of UAV landslide images after the Wenchuan earthquake, and successfully applied it to the Haiti earthquake landslide; however, the model parameters are too many and the computer performance requirements are too high. Kubo et al. [37] aimed to explore the impact of different image enhancement modes on detection accuracy. The study found that when the shape of the slope damage area is maintained and the image data are enhanced, the result is better than that without enhancement. Recently, Liu et al [38] proposed a Mask R-CNN+++ model using interferometric synthetic aperture radar (InSAR) images of the eastern edge of the Tibetan Plateau, which can identify landslides with high accuracy. However, the blurred and discontinuous shape of the deformed patches in the images leads to small-scale deformed objects being easily overlooked..

In summary, there are still some problems in the application of deep learning algorithms for landslide identification, such as how to achieve accurate identification on a small number of landslide datasets and how to fully learn the semantic information of landslides. Therefore, the purpose of this paper is to develop a better model for extracting landslide information captured by UAV using the well-structured instance segmentation model Mask R-CNN, and to make data support for the subsequent study of landslide influencing factors in the area. In this paper, the following improvements are made to Mask R-CNN: (1) addition of the CBAM attention mechanism to the backbone network ResNet to enable it to better extract landslide information. (2) A bottom-up channel is added to the feature pyramid network to incorporate the semantic features of each channel more adequately. (3) Replacement of the traditional RPN with the GA-RPN to guide the generation of anchors and to reduce redundant anchors.

## 2. Materials and Methods

### 2.1. The Mask R-CNN Model

Mask R-CNN is an instance segmentation model proposed by Kaiming He et al. [39], which can be used to accomplish a variety of tasks, including target classification, target detection, semantic segmentation, instance segmentation, and human pose recognition, and which shows its easy-to-use features to the fullest. The whole idea of the Mask R-CNN algorithm is simple. The authors follow the classical target detection algorithm Faster R-CNN and add the classical semantic segmentation algorithm fully convolutional network (FCN) [40] to generate the corresponding Mask branches. Mask R-CNN uses a two-step structure like that of Faster R-CNN, where the first stage is a RPN that extracts the candidate target bounding boxes, and the second stage is a Mask R-CNN model that extracts features from the candidate regions of the RPN using Region of Interest Alignment (RoI Align) and performs category classification, bounding box regression and binary mask generation. The features applied in both phases can be shared for faster inference. As the pixel bias problem in RoI pooling is identified, a corresponding RoI Align strategy is proposed. RoI Align avoids the error introduced by two-coordinate ‘quantization’ rounding by bilinear interpolation, and the mask prediction branch predicts the object mask so that Mask R-CNN can obtain high accuracy. 

As shown in Figure 1, the steps of the Mask R-CNN algorithm are presented as follows: first, images such as landslides are input to the neural network ResNet [41] to obtain multiscale feature maps corresponding to each stage of $C_{2}$–$C_{4}$, and then the feature maps of different scales are unsampled by a series of convolution in the FPN [42] and fused with the corresponding feature maps of ResNet to form the feature maps of stages $P_{2}–P_{5}$ The $P_{6}$ stage is obtained by direct downsampling from the $P_{5}$ stage. Considering that the $C_{1}$ layer in ResNet is too large and contains too many parameters, it is not involved in the construction of the FPN. Numerous RoI candidates are obtained by traversing all feature maps to generate predetermined anchors at each point. These candidates are conveyed to the RPN network for binary classification (SoftMax function to classify foreground or background) and Bounding Box regression, after which some RoI candidates are filtered and the remaining candidates are subjected to the RoI Align operation. These candidates are input to the fully connected layers (FC layers) for multicategory classification, bounding box regression, and mask prediction in the FCN.

#### Loss Function

Mask R-CNN is a re-addition of split branches on top of the Faster R-CNN architecture, the network structure has changed, and the final loss function has changed accordingly. Due to the addition of mask branches, the loss function of each RoI is shown below:(1)$$L=L_{cls}+L_{box}+L_{mask}$$

There is no difference between the loss function $L_{\mathrm{cls}}$ and $L_{box}$ that is defined in Faster R-CNN. Regarding $L_{mask}$, assuming a total of K categories, the output dimension of the mask segmentation branch is K × m × m, and for each point in m × m, K binary masks are output (using the sigmoid output for each category). Note that when calculating the loss, the binary cross-entropy loss is not calculated for each category of sigmoid output, but the loss is calculated for the sigmoid output of the category to which the pixel belongs. In the experiment, this study selects the corresponding mask prediction by the category of the classification branch prediction. In this way, the mask prediction and t classification prediction are completely decoupled.

### 2.2. Structure of the Improved Mask R-CNN

Although Mask R-CNN is the most classical and advanced pixel-level instance segmentation model, the recognition effect on this study area has the following problems: (1) the distribution, shape, size, and texture of landslides vary greatly, and ResNet convolution may ignore small landslide features. (2) Although FPN extracts multi-scale features in the image, the features input to the RPN layer only contain the feature letter information of this layer and the previous layer, and there is a loss of feature information in the remaining layers. (3) The anchor in RPN is fixed size, fixed scale, and fixed number, and the sample labeling of different size targets will have errors, which affects the detection accuracy. Therefore, the following improvements are made to the Mask R-CNN framework in this study.

#### 2.2.1. Attention Mechanism Added to the Backbone Network ResNet

This paper observes that the texture features of landslide and non-landslide texture features in the dataset, such as the surrounding bare land and farmland, are easily confused. Thus, an attention mechanism is added to the 1 × 1 convolution in the FPN structure after extracting features in the $C_{2}–C_{4}$ stages of ResNet to strengthen the information extraction of the network for landslide features. The attention mechanism is essentially a mechanism that focuses on local information, which can locate the information of interest in the image and suppress useless information, and the results are usually presented in the form of probability maps or probability feature vectors. In this paper, the CBAM proposed in the literature [43], which is a hybrid spatial and channel attention mechanism, is adopted. 

In terms of implementation, first, the input feature map is modeled with channel-dimensional attention, and the importance of different channels is learned by applying corresponding attention weights to each feature channel. Second, the feature map is modeled with spatial–dimensional attention, and the target of interest is focused on the spatial location by applying corresponding attention weights to different spatial locations of the feature map. Introducing the CBAM attention mechanism enables the Mask R-CNN model to better focus on the most important feature channels and spatial locations in the image during the training and prediction process, as indicated in Figure 2.

#### 2.2.2. The Structure of Improved FPN

Although the FPN structure performs well in many computer vision tasks, it usually contains only semantic information in the higher-level feature maps, while the lower-level feature maps contain more detailed information. The feature maps input to the RPN contain only the feature information of the current and upper layers, and lack the lower-level feature information, which is equally important for detection tasks.

Based on the above analysis, an improved FPN is developed in this study by adding channels that are connected backward from the bottom to the top, and the additional channels can help the network extract feature information at different levels, as shown in Figure 3, where $P_{2}$,$P_{3},P_{4}$,$P_{5}$, and $P_{6}$ are feature layers of the FPN. The newly added bottom-up path merges the low-level feature map $N_{i}$ and the high-level feature map $P_{i+1}$ to generate a new feature map $N_{i+1}$. The specific operation is described as follows: First, the feature map size of $N_{i}$ is reduced by downsampling the 3 × 3 convolution of $N_{i}$ with stride = 2 to obtain a feature map with the same size as $P_{i+1}$. Second, each element of the feature map $P_{i+1}$ and each element of the downsampled feature map are added using the lateral join, and the new feature map $N_{i+1}$ is generated by processing the new feature map with a 3 × 3 convolutional layer of stride = 1. The specific operation is shown in Equation (2) and Figure 3.
(2)$$N_{i+1}=\left(N_{i}\times Conv_{size=3}^{stride=2}+P_{i+1}\right)\times Conv_{size=3}^{stride=1}$$

The newly generated feature maps $N_{2}$,$N_{3}$,$N_{4}$,$N_{5}$, and $N_{6}$ use different levels of feature maps to extract feature information at different scales, while avoiding excessive information loss, helping the network to better handle targets of different sizes and improve the accuracy of the detection task.

#### 2.2.3. The Improved RPN

Since Wang et al. [44] suggest that the design of the anchor should not rely entirely on manual experience to set the number, scale, and proportion, first, an unreasonable setting will affect the detection results. Second, the anchor should not be set too densely, resulting in a very large computation. Last, the ratio of positive and negative samples of the anchor can easily be unbalanced, so there should be guidance in designing the anchor. GA-RPN was developed to generate anchors in a guided manner by using key points (center points) to predict anchor positions and sizes and then by refining the anchors.

When guiding the generation of anchors, two parts are included: anchor location prediction and anchor shape prediction. In anchor location prediction, the possibility of each point of the feature map being used as the object center is predicted, and these response areas will be used as anchor centers. Since the location response is performed, the background can be eliminated, and thus, the number of anchors to be placed can be greatly reduced. The background is not considered in the inference stage, and after predicting the position, the normal convolution is employed instead of mask convolution, which is computed only where there is an anchor for acceleration.

After obtaining the anchor position, the shape of the anchor is then determined. This branch learns directly from the original image and directly learns the width w and height h information of each pixel point. The width and height are obtained by using dw and dh mapping because the range of both values is too large, which can cause instability in prediction.

The authors also designed the feature adaptation module to solve the feature-matching problem. The module incorporates the shape information of the anchor directly into the feature map so that the newly obtained feature map can adapt to the shape of the anchor at each location.

The loss function is added to the original $L_{cls}$ and $L_{reg}$ of the RPN with the functions of position $L_{loc}$ and shape $L_{shape}$.
(3)$$L=\lambda_{1}L_{loc}+\lambda_{2}L_{shape}+L_{cls}+L_{reg}$$

The guided anchors generated by the above enhancements not only enhance the quality of anchors but also avoid many redundant anchors in the RPN, which accelerates the model training and enhances the model accuracy. This paper replaces the original RPN network with GA-RPN, as shown in Figure 4.

### 2.3. Flowchart of Landslide Detection

The flowchart of this paper is shown in Figure 5. First, the high-resolution landslide images taken by the UAV are acquired, and then the data are processed. The orange box shows the processing: cropping, labeling, and data enhancement. Second, the data are input into different models for training, testing, and extracting landslide information. Last, the results are analyzed and evaluated using metrics.

## 3. Experiments

### 3.1. Dataset Source

This experimental study area is in Sanming City, Fujian Province, where mountains and hills account for more than 80% of the total area of the province. It is located between the Wuyi Mountains and the Daiyun Mountains. Therefore, it is particularly prone to geological disasters such as landslides, flash floods, and mudslides during periods of concentrated rainfall. 

Figure 6 shows the monitoring area range and landslide distribution. Sanming had a landslide caused by heavy rainfall on 5 June 2019. UAV data were acquired on 12 June 2019, with the following specific indicators in Table 1. The landslides in the study area were mainly captured after heavy rainfall. According to Varnes’s classification [45,46], the main types of landslides are debris flow and debris avalanche inflows, and the main factors of landslide generation are strong surface water flow erosion and movement of loose soil or rocks on steep slopes due to heavy precipitation. Other causes of landslide formation may be weak or sensitive material ground cover in geological causes, excavation of slope or its toe, loading of slope or its crest, deforestation, irrigation in human causes, etc. 

The total area of landslides in the study area is 316,738.58 $m^{2}$, with an average area of 1314.28 $m^{2}$. According to the interpretation results of UAV images, the maximum statistical disaster coverage area is 14,289.63 $m^{2}$, and the minimum coverage area is 14.76 $m^{2}$.

The landslides were visually interpreted by experts and verified in the field to ensure a final interpretation accuracy of 97%, which can be used as a reference map for landslides, and a total of 241 landslides were mapped. 

After obtaining the visual interpretation samples, we classified the samples according to the area of the landslide. The Descriptor column of Table 2 divides the landslides into 6 categories. Landslide area less than 200$\;m^{2}$ is labeled as Very small, area between 200 and 2000 is defined as Small, area between 2000 and 20,000 is Medium, area between 20,000 and 200,000 is Large category, area between 200,000 and 2,000,000 is Very large, and Huge for areas larger than 2,000,000. Subsequently, we count the number of landslides at different levels in the study area; 61 landslides belong to the Very small category, 136 to the Small category, and 44 to the Medium category. There is no distribution of landslides in the large and above category.

Referring to the classification levels in Table 2, the landslides are classified as shown in the legend on the right side of Figure 7, and the unit is $m^{2}$.

### 3.2. Dataset Preparation

Considering the large size of the remote sensing images captured by the UAV, the model has a limitation on the input data size, so the images need to be cropped. To better reflect the texture features of the landslide and the recognition of the model, counting the area of all sample labels, according to the area distribution, we fix the size of the input to 256 × 256. Figure 8 shows a similar cropping process to cut the original image as well as the labeled images and construct the landslide dataset according to the requirements of the model. All the samples are divided into the training set, validation set, and test set in the ratio of 6:2:2, i.e., 1050 images are adopted as training samples, 300 images were used as validation sets to verify the optimized hyperparameters, and the remaining 300 images were used to test the model effect. The data enhancement technique allows the landslides to show different orientations, structures, and boundary shapes, thus increasing the size and diversity of the data [37]. Therefore, we enhance the training samples by flipping them up and down, left and right, etc. Finally, a total of 4050 landslide samples are obtained for model training.

### 3.3. Experimental Procedure

The experimental hardware configuration is expressed as follows: processor Intel (R) Core (TM) i7-8700K processor, running memory of 12 GB, graphics card model NVIDIA RTX3080Ti, and deep learning framework for PyTorch, mainly on the MMDetection tool library to implement Mask R-CNN model training. MMDetection [47] is a PyTorch-based target detection tool library developed by open-mmlab, and MMDetection v2.19.0 is chosen for this experiment. The experimental data label name is set to landslide, 50 epochs are set, the initial learning rate is set to 0.005 and it was reduced by a ratio of 0.1 after the 30th and 40th epochs, the weight decay factor is 0.0001, and the momentum factor is 0.9. In this paper, the pretraining weights obtained by Mask R-CNN on the official COCO2017 [48] dataset are used as the migration learning method. 

### 3.4. Evaluating Indicator 

Precision, Recall, and Accuracy are used to quantitatively evaluate the extraction results of the model. Precision measures the proportion of correctly extracted landslides to the total number of extracted landslides [49]. Recall is the proportion of all correctly extracted slopes. Accuracy measures the overall correctness of the model’s predictions. The formulas for precision, recall, and accuracy are expressed as follows:(4)$$\mathrm{Precision}=\frac{\mathrm{TP}}{\mathrm{FP}+\mathrm{TP}}$$
(5)$$\mathrm{Recall}=\frac{\mathrm{TP}}{\mathrm{FN}+\mathrm{TP}}$$
(6)$$\mathrm{Accuracy}=\frac{\mathrm{TP}+\mathrm{TN}}{\mathrm{TP}+\mathrm{FN}+\mathrm{FP}+\mathrm{TN}}$$
where TP means the number of positive cases where the model predicts correctly; FP means the number of positive examples of model prediction errors; FN means the number of negative cases of model prediction errors; and TN means the number of negative cases correctly predicted by the model.

Generally, increasing precision often leads to a decrease in recall and vice versa. To comprehensively measure the goodness of the detection model, the F1 score (F1) is introduced to evaluate the model. F1 is a method that combines precision and recall to evaluate the performance of a classification or detection model. The higher the F1 value, the better the performance of the model; the formula is illustrated as follows.
(7)$$F1=2\times \frac{\mathrm{Precission}\times \mathrm{Recall}}{\mathrm{Precission}+\mathrm{Recall}}$$

In this paper, landslide shape segmentation is simultaneously realized, and the mean intersection over union [50] (MIoU) is introduced to reasonably evaluate the segmentation results. MIoU calculates the ratio of the intersection and union of two sets, which in this paper are landslide interpretation maps and landslide prediction maps, respectively. The greater the MIoU ratio is, the higher the accuracy ratio; its formula is:(8)$$\mathrm{MIoU}=\frac{1}{n+1}\overset{n}{\underset{i=0}{\sum }}\frac{P_{ii}}{\sum_{j=0}^{n}P_{ij}+\sum_{j=0}^{n}P_{ji}-P_{ii}}=\frac{\mathrm{TP}}{\mathrm{FP}+\mathrm{TP}+\mathrm{FN}}$$
where n is the number of predicted categories, $P_{ii}$ denotes the original category i predicted to be category i, $P_{ij}$ denotes the original category i predicted to be category j, and $P_{ji}$ denotes the original category j predicted to be category i. 

## 4. Results

### 4.1. Extraction Accuracy of the Proposed Improved Mask-R-CNN Model

The performance of several models based on Mask R-CNN was quantitatively analyzed by the distribution of statistical confusion matrices, and Table 3 summarizes the five metrics regarding the accuracy, Recall, Accuracy, F1, and MIoU of landslide extraction. The results are retained to one decimal place. The improved Mask R-CNN model in this paper obtained the highest scores in all metrics, Precision (93.9%), Recall (91.4%), Accuracy and F1 scores were the same (92.6%), and MIoU (86.4%).

The original Mask R-CNN model obtained the lowest scores for Precision (86.9%), Recall (78.5%), Accuracy (81.7%), F1 (82.5%), and MIOU (70.2%). The differences in the metrics between the two models (the original model and the improved model) are 7%, 12.9%, 10.9%, 10.1%, and 16.2%, which indicates that the improved Mask R-CNN model in this paper is significantly better in landslide detection and that the proposed model in this paper is feasible. 

### 4.2. Visualization Comparison of Landslide Extraction Results

Figure 9 shows the results of different improved modules to extract the same set of landslide contours. This set of images is selected from a sample of landslides at different locations with variations in lighting and shading.

Comparing the extraction results of each method, Figure 9(Aa) shows a single landslide with a larger area labelled, and Figure 9(Ba–Da) labels the case where two landslides are present in the sample. The original model has a more serious case of wrong detection and omission for this group of landslides.

As the first module modified in this paper, Figure 9(Ac–Dc) adds CBAM to the ResNet backbone, and we find that CBAM has the same effect in ResNet as in [25], where the network focuses more on landslide area features, as shown in Figure 9(Ac), the false detection of surrounding vegetation is reduced. Figure 9(Bc) filters out the small green vegetation in the middle, although the small landslide in the upper right fails to be identified. The confusion of landslide labels below Figure 9(Cc) has also been reduced and it even detects a small area of landslide.

In this experiment, GA-RPN was used instead of RPN to take into account that non-fixed anchors are more favorable for the variability of landslides of different sizes. Figure 9(Bd) shows that in the Mask R-CNN model with GA-RPN, the quality of anchors is high and the model identification is accurate. Although, in this model, a small area of landslide cannot be recognized after layer-wise convolution and semantic information is ignored. However, in Figure 9(Dd), the surrounding environment is relatively uncomplicated and small-area landslides are successfully recognized.

Adding a bottom-up channel to the existing structure of the FPN, it was shown that this channel can adequately integrate low-level landslide location information and high-level landslide semantic information as shown in [34]. The richer the landslide information learned by the model, the more accurately the anchor points are located, and the better the fit of the mask after full convolution. Figure 9(Be) successfully discriminates between large and small landslide contours, although there are still errors in contour information extraction. The landslide below Figure 9(Ce) is disturbed by vegetation, and the improved FPN+Mask R-CNN model cannot fully learn the features of this landslide.

The improvement of a single module does not solve the problem of various types of landslides in the dataset well: the edge information of large landslides, the identification of small landslides, the interference of vegetation, bare soil, living areas and other environments, etc., all add difficulties to the accurate identification of landslides [51]. Therefore, in this paper, the three modules are fused, and Figure 9(Af–Df) shows the effect of the improved model recognition. Both the single landslide in Figure 9(Af), with accurate landslide location and correct shape, and the landslide in Figure 9(Bf), with both large and small landslides, are successfully recognized, and the anchor box is completely correct. Such a phenomenon is also clearly reflected in Figure 9(Cf), where the underside of the landslide is clearly marked and the boundary matches the GT map; in Figure 9(Df), the boundary information of the small landslide is also the most accurate among all methods.

By comparing these five sets of detection results, we find that adding CBAM, which focuses more on channels and space, can focus more on the features of the landslide in the sample and mitigate the background interference; replacing RPN with GA-RPN can autonomously learn the shape of landslides for feature adaptation; adding bottom-up channels in the FPN structure combines the bottom location information with the top semantic information, which is more conducive to landslide information extraction. 

We judge that it is necessary to fuse the three modules in order to fully learn the landslide boundary contour features and to generate anchor points flexibly according to the region of interest, so we propose an improved Mask R-CNN model that fuses the three modules, and the experimental results show that the model has the best landslide extraction effect.

In terms of model detection speed, Table 4 (with four decimal places retained) shows that the speed difference is not significant at higher accuracy, and for the same epoch, the time required for each iteration differs by 0.0833 s, indicating that the model in this paper achieves high accuracy and without much time loss.

### 4.3. Landslide Extraction Results for Different Area Scales

Figure 10 shows the extraction performance of the improved model for different distribution cases. The first row shows multiple landslides in the sample, and the second, third, and fourth rows show the individual landslide in the sample, where each row has a different level of landslide area. The fifth row shows the landslides sample that is not identified by the improved model.

Compared with the GT maps in the pink box, the improved model can accurately identify pixels related to landslides and label them as landslide areas. The accuracy of landslide identification for multiple or single landslides is basically above 0.94, and the identification rate of small landslides in the fourth row is also above 0.76. The reason why the accuracy of Figure 10(i5,i8) is lower may be that the landslide area is too small, and the model failed to extract the shape of the landslide well compared with the corresponding GT map. The reasons for missed detection in the fifth row may be due to the influence of sunlight and shadows on the image, as well as interference from surrounding residential and construction sites, such as Figure 10(j8) being obscured by shadows. By observing the dataset, we found that the relatively small number of landslide samples around buildings prevented the model from learning adequately, resulting in missed samples accounting for about 6.5% of the test set [52].

To further explore the improved model for landslide detection of different gradations, this experiment inputs the cropped images of the study area into the improved Mask R-CNN model for detection, imports the recognition results into ArcGIS software for raster data mosaic, keeps the parameters of pixel type, projection, and raster cell size of the raster data the same as the base image, and converts the raster data into vector polygons. The area of each landslide polygon after mosaicking was calculated to investigate the effectiveness of the improved model in identifying very small, small, and medium categories. The improved model identified 241 landslides successfully, but there were differences in the area distribution. Table 5 provides statistics on the relevant data (the results are retained to two decimal places), where the Total column is the total area of landslides, the total area detected by the improved model is 19,430.4 $m^{2}$, 6.13% less than the actual area, and the overall detection accuracy of the model is 93.87%. The detection accuracy of the model for three different areas was 93.62%, 96.29%, and 92.90%. The model achieved the highest extraction accuracy in small grade, and slightly decreases in very small grade and medium grade, indicating that the model is more sensitive to the detection of landslides in small areas. 

Figure 11 shows the visualization results of landslide extraction in part of the study area. The legend in the lower right corner indicates the labelled landslide contour, while the blue-filled area is the landslide detected by the improved model. The figure on the right shows examples of landslide extraction results for three levels. The orange circles indicate the areas that were not recognized by the model. For the medium level, there were many missed parts in the two landslides, resulting in the lowest detection accuracy. For the small level, the two recognized landslides almost completely overlapped with the annotated areas, indicating that the improved model has a credible performance in extracting small landslides.

## 5. Discussion

### 5.1. Comparison of the Training Loss at Different Backbones

Comparing the training loss of ResNet at different network depths, four models, ResNet101, ResNet101+CBAM, ResNet50+CBAM, and ResNet50+CBAM+Improved FPN+GA-RPN, were trained based on the same dataset, and Figure 12 shows their training loss curves. It can be seen that CBAM enables the network to better capture the spatial information of the image to reduce the loss faster in training, thus improving the performance of the model. Meanwhile, ResNet50+CBAM reduces the number of parameters by half than ResNet101+CBAM, and the loss decreases a little faster. Therefore, ResNet50 is chosen as a suitable backbone for feature extraction.

### 5.2. Comparing Effectiveness with Other Detection Models

In Section 4.2, we have compared the effectiveness of the initial Mask R-CNN model with our improved model for landslide detection. Table 6 demonstrates that the improved Mask R-CNN model has high detection accuracy and takes less time. However, in addition to the Mask R-CNN model, there are several other deep network models that can be used for landslide extraction. Therefore, we also did experimental comparisons with other deep learning detection models based on the same dataset.

Table 6 shows the relevant metrics, and the results are kept to one decimal place. The parameters of the Faster R-CNN model, such as learning rate, weight decay, etc., are consistent with those of the model in Table 3. The SSD model used the same parameters for experiments but the gradient explosion occurred, and after the study, we adjusted the warmup learning rate iteration of the model to 2000, and the rest parameters were kept the same; we also trained the YOLOv3 [53], while the effect was not very satisfactory under the same parameters, considering the principle of comparison under consistent conditions, so we did not continue to explore the results of tuning the parameters; hence, the index results of the YOLOv3 model are not listed. According to Table 6, it can be found that the accuracy of Faster R-CNN is higher than SSD, but none of them is as accurate as the original Mask R-CNN model detection, not to mention the recognition effect of the improved model proposed in this paper. As stated in Section 2.1, Mask R-CNN makes the detection efficiency higher than Faster R-CNN because of the innovative proposed RoI Align and mask branches. The one-stage model SSD and YOLOv3 use only one convolutional neural network to locate and classify all targets directly on the whole image, skipping the step of generating candidate regions, so the model detection is fast; however, the regression algorithm makes the model detection accuracy low, especially when the target size and aspect ratio vary greatly.

### 5.3. Advantages and Disadvantages of the Improved Model for Landsilde Extraction

In this study, we add the CBAM attention mechanism to ResNet, and the impact of the attention mechanism has been widely studied; for example, Sui et al. [30] used to detect the damage of post-earthquake buildings, and the experiments show that CBAM can indeed pay more attention to important channel information. Liu et al. [38] replaced backbone by ResNext network [54], and discussed the impact of two attention mechanisms, CBAM and shuffle attention (SA) [55]; the results show that the inclusion of the attention mechanism is more effective. Liu et al. [34] also replaced the backbone with ResNext, and they demonstrated that adding a bottom-up channel to the FPN did result in better fusion of feature information as well.

All the above studies have achieved positive results, but they are all based on post-earthquake landslide detection. This paper focuses on landslide collapse caused by strong rainfall. Taking Sanming City as the study case, more than 80% of the landslide area is less than 2000 $m^{2}$; how to quickly identify the distribution of small landslides due to strong rainfall captured by UAV is the goal of this paper. We use data enhancement techniques to expand the size of the training set, then we explored the effect of models with different backbone depths using the training loss curve as an indicator under the same conditions, and selected ResNet50+CBAM with faster descent speed as the backbone network of the model. We also selected the GA-RPN network that can adaptively choose the optimal anchor size and aspect ratio to adapt to different sizes and shapes of landslides to generate an efficient RoI and reduce redundancy. The extraction of landslides in Sanming City was performed with 92.6% accuracy, which can meet the requirements of landslide detection.

In this study, we only selected the visible band of the UAV as the input data, and the experimental results show that the interference from the background noise of the surrounding features cannot be completely eliminated. Referring to other landslide detection studies [56], fusing more data from other regions can improve the accuracy of the model detection. In subsequent studies, we can train and validate by accessing additional landslide datasets to improve the model and increase its generalization to new test areas. In addition to remotely sensed images, other data, such as digital elevation model (DEM) data, can also be incorporated into the model [57].

The experimental results of this paper show that the depth and parameters of the deep learning network have a great influence on the accuracy of the extraction results. However, the improvement and optimization of the deep network model is a challenging task. For example, Al-Zuhairi et al. [58] aimed to explore the most suitable model architecture with the best parameters. Cheng et al. [26] discussed the effect of the location of the attention mechanism on the model performance and the comparison with other attention modules. Different training techniques can lead to different features of the model, resulting in biased prediction results of the model. In addition, different data enhancement methods, activation functions, and loss functions can also have different effects on the model’s accuracy. In this paper, we choose the Mask R-CNN standard parameters; the next step is to explore the best combination of parameters affecting the performance of the model by adjusting these parameters for training. 

## 6. Conclusions

Due to its geographical location and mountainous structure, Sanming City in Fujian Province is a prone area to landslides. UAVs are highly effective in obtaining landslide information due to their flexibility and low cost, but their high resolution also brings background noise interference. Manual extraction is time-consuming and has low efficiency. In contrast, the Mask R-CNN model has strong robustness to extract landslides. 

In the study, we annotate the UAV images of Sanming City and establish an unmanned aerial landslide sample dataset. Then, we propose an improved model based on the Mask R-CNN model to address the problems of recognizing landslides in the study area: (1) the ResNet50 convolution is prone to missing small landslides in the samples, so we added the CBAM mechanism to focus more on landslide features; (2) the FPN feature layer adds bottom-up channels to fully integrate bottom-level semantic information and top-level location information; (3) we notice that the original model’s anchor could cause labelling errors in this dataset, so we replace it with the advanced GA-RPN network. We also use data augmentation techniques and transfer learning methods and selected ResNet50+CBAM, which had a faster decrease in training loss, as the backbone network for the improved model. The improved model aimed to focus more on landslide features and achieve higher accuracy in extracting landslide information. The experimental results show that the improved model effectively reduced the problems of the original model, with an accuracy of 93.9%, a recall rate of 91.4%, and an accuracy of 92.6%. Compared with the original Mask R-CNN model, these parameters increased by 7%, 12.9%, and 10.9%, respectively.

We also evaluate the improved model’s landslide recognition effect at different levels and found that the highest detection accuracy is a small level, which is 96.29%. This finding indicates that the model’s ability to recognize small landslides caused by heavy rainfall is satisfactory, but the robustness of the model on landslide extraction in other areas is worth exploring.

In the future, the effectiveness of the improved model in terms of generality and portability can be explored by expanding the dataset and adding different data sources, including satellite images and UAV images with different resolutions. The improvement of landslide recognition accuracy in Sanming City provides a more accurate landslide area for the study of its influencing factors and is conducive to exploring the factors affecting landslide occurrence frequency, providing reference significance for the study of landslide disasters in other areas.

## Figures and Tables

**Figure 1 sensors-23-04287-f001:**
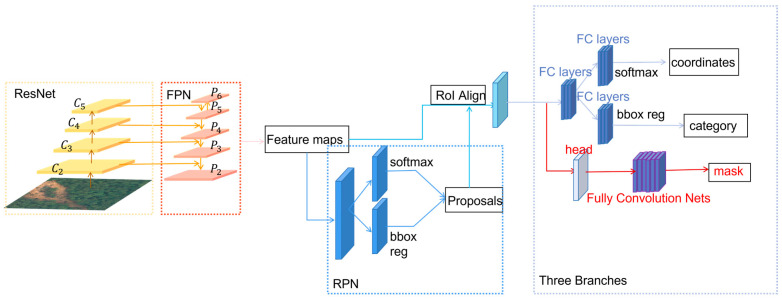
The structure of Mask R-CNN.

**Figure 2 sensors-23-04287-f002:**
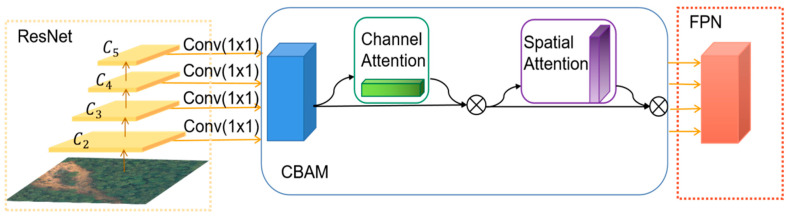
ResNet backbone with CBAM.

**Figure 3 sensors-23-04287-f003:**
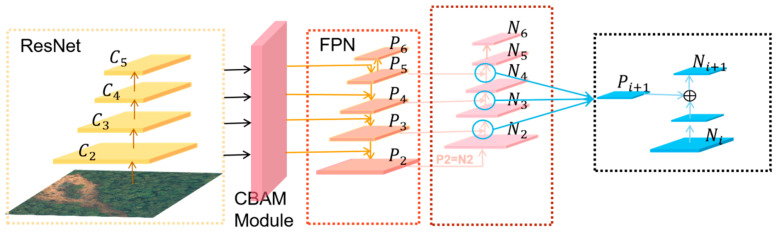
The FPN structure with bottom-to-top channels.

**Figure 4 sensors-23-04287-f004:**
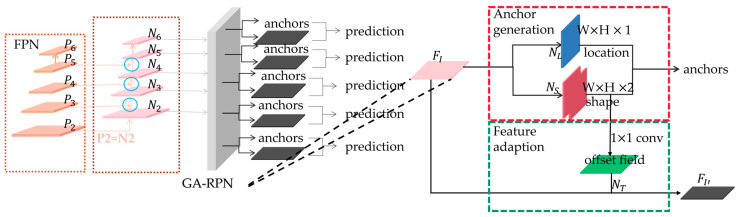
Replace the RPN structure with GA-RPN.

**Figure 5 sensors-23-04287-f005:**
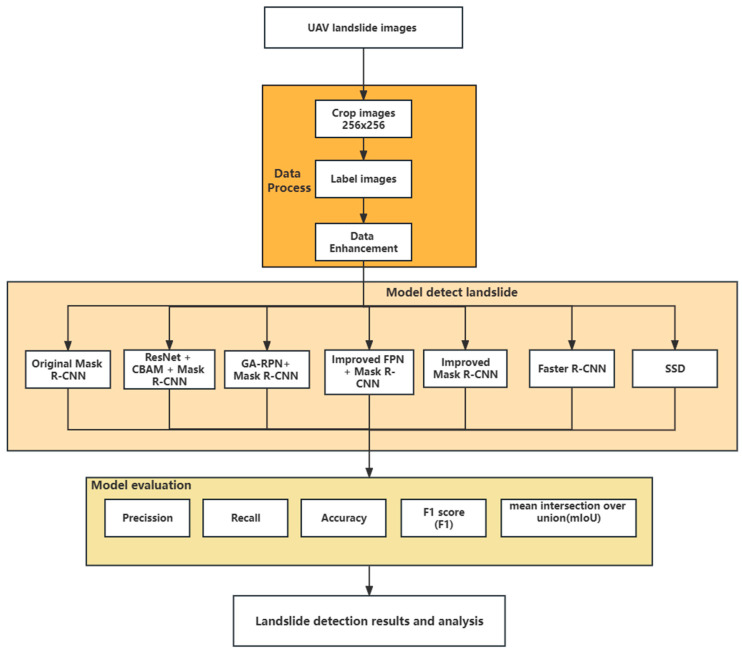
The flowchart of landslide detection in UAV image.

**Figure 6 sensors-23-04287-f006:**
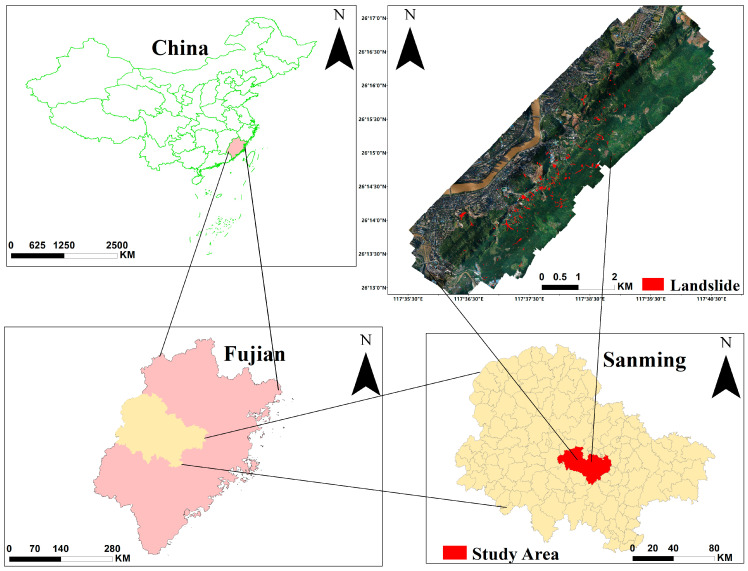
Study Area.

**Figure 7 sensors-23-04287-f007:**
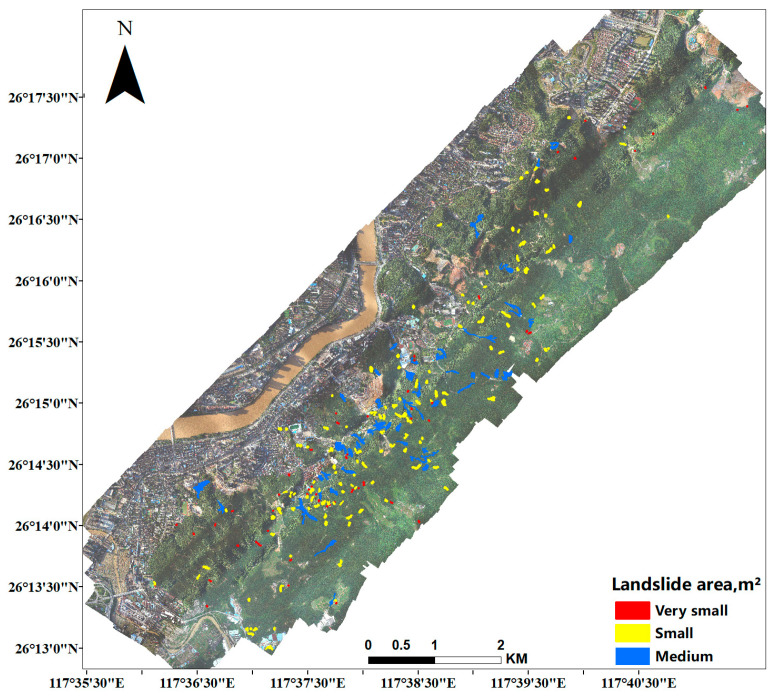
Landslide distribution with different levels of the study area.

**Figure 8 sensors-23-04287-f008:**
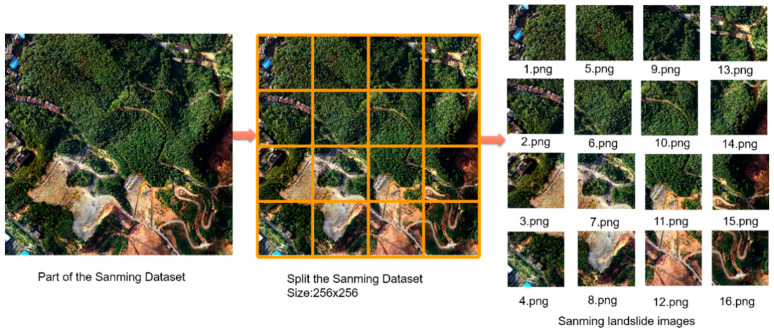
Preparation of the Sanming dataset.

**Figure 9 sensors-23-04287-f009:**
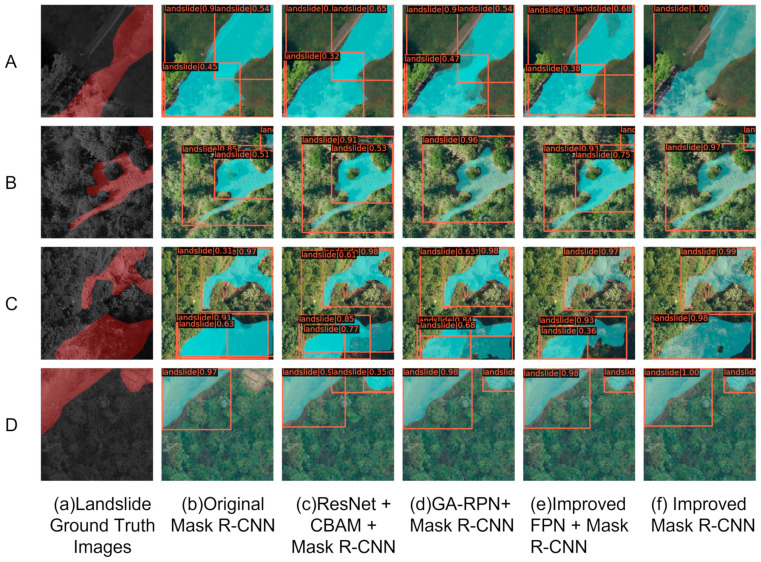
Comparison results of several methods. (**Aa**–**Da**) shows the GT images; (**Ab**–**Db**) represents original model detection results. (**Ac**–**Dc**) shows the predictions of the model with CBAM added to ResNet. (**Ad**–**Dd**) represents detection results of the model with GA-RPN replacing RPN; (**Ae**–**De**) shows the predictions of the model with improved FPN; (**Af**–**Df**) shows the detection results of an improved model of the fusion of the three modules proposed.

**Figure 10 sensors-23-04287-f010:**
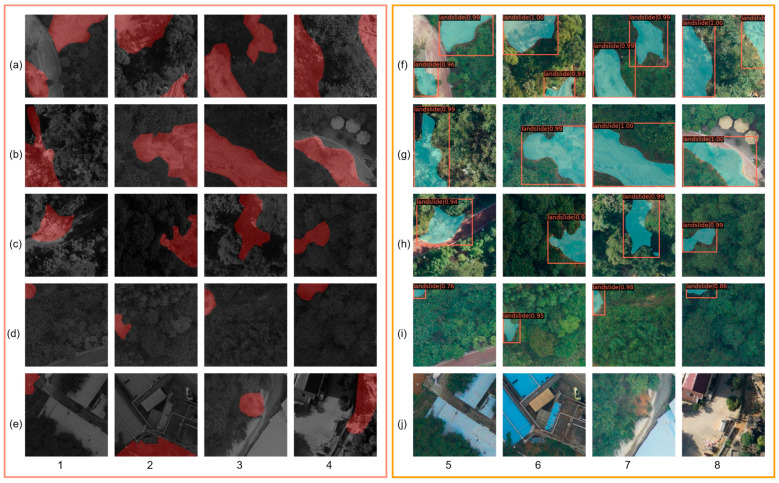
Extraction results. (**a1**–**e4**) in the pink box are the GT maps of the landslides; (**f5**–**j8**) in the orange box are the corresponding maps of the landslide results identified by the improved Mask R-CNN.

**Figure 11 sensors-23-04287-f011:**
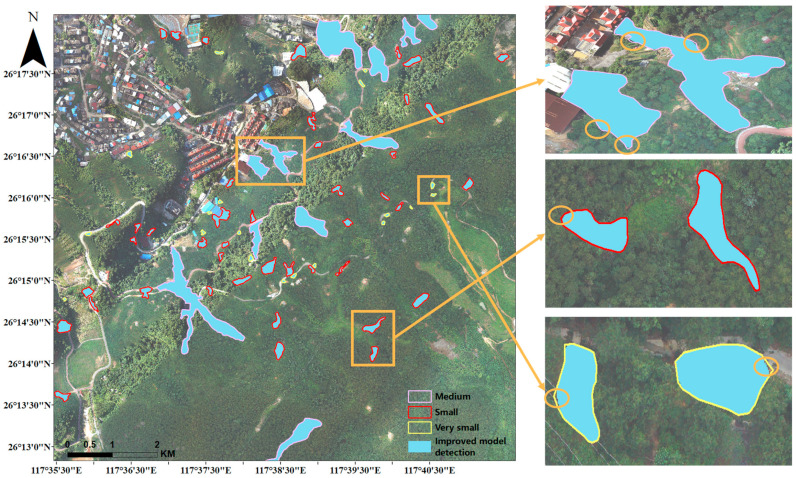
The effect of landslide extraction at different gradations.

**Figure 12 sensors-23-04287-f012:**
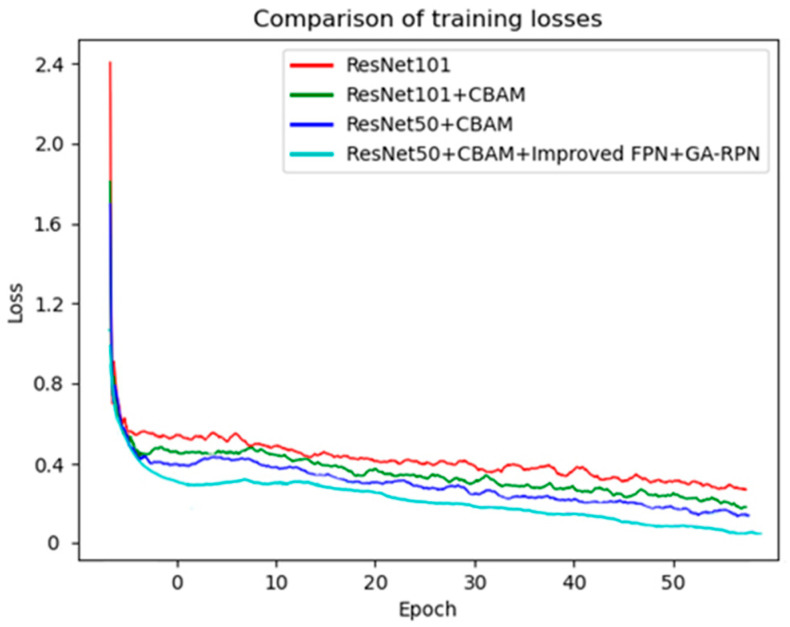
Training losses of different models.

**Table 1 sensors-23-04287-t001:** Original Image Parameters.

Image Parameters	Value
Resolution	0.1 m
Row	85,406
Column	84,370
Number of Bands	3 (Red, Green, Blue)

**Table 2 sensors-23-04287-t002:** Number of landslides at different levels in the study area.

Descriptor	$\mathrm{Area}/m^{2}$	Number
Very small	<200	61
Small	200–2000	136
Medium	2000–20,000	44
Large	20,000–200,000	0
Very large	200,000–2,000,000	0
Huge	>2,000,000	0

**Table 3 sensors-23-04287-t003:** Performance metrics for several models based on Mask R-CNN. The bold is to to highlight the most accuracy between models.

Model	Precision/%	Recall/%	Accuracy/%	F1/%	MIoU/%
Original Mask R-CNN	86.9	78.5	81.7	82.5	70.2
ResNet+CBAM+Mask R-CNN	87.1	80.5	83.1	83.7	71.9
GA-RPN+Mask R-CNN	89.3	84.6	86.6	86.9	76.8
Improved FPN+Mask R-CNN	90.6	87	88.6	88.7	79.8
**Our Improved Mask R-CNN**	**93.9**	**91.4**	**92.6**	**92.6**	**86.4**

**Table 4 sensors-23-04287-t004:** Comparison of model detection speed.

	Model Detection Speed
Improved Mask R-CNN	0.2493 s/iteration
Original Mask R-CNN	0.1660 s/iteration

**Table 5 sensors-23-04287-t005:** Landslide detection area for different gradations.

Descriptor	Actual Area/m^2^	Improved Mask R-CNN Detection Area/m^2^	Detection Accuracy
Total	316,738.58	297,308.18	93.87%
Very small	6379.94	5972.71	93.62%
Small	88,775.84	85,485.81	96.29%
Medium	221,582.79	205,849.66	92.90%

**Table 6 sensors-23-04287-t006:** Performance metrics for several other models. The bold is to highlight the most accuracy between models.

Model	Precision/%	Recall/%	Accuracy/%	F1/%	MIoU/%
Faster R-CNN	77.8	71	77.3	74.4	59.1
SSD	72.7	67.5	69	70	53.9
Mask R-CNN	86.9	78.5	81.7	82.5	70.2
**Our Improved Mask R-CNN**	**93.9**	**91.4**	**92.6**	**92.6**	**86.4**

## Data Availability

Not applicable.

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
