# Peer review of "Enhance the Accuracy of Landslide Detection in UAV Images Using an Improved Mask R-CNN Model: A Case Study of Sanming, China"

_sensors, 2023, doi:10.3390/s23094287_

Round 1
Reviewer 1 Report
This manuscript presents an interesting approach to landslide detection using an improved version of the Mask R-CNN model. The authors focus on the specific context of Sanming City in Fujian Province, a landslide-prone area. The study demonstrates the effectiveness of the improved model and provides valuable insights for future research in this area. However, obscure issues still exist that should be addressed before accepting this version.
GENERAL COMMENTS
1. The study focuses on a single location, Sanming City, which limits the generalizability of the findings to other landslide-prone areas. An improved strategy is to add a research area to the title. A candidate title would be “Enhance the accuracy of landslide detection in UAV images using an improved Mask R-CNN model: a case study of Sanming, China.”
2. The authors acknowledge the limitations of the model in recognizing small landslides and the impact of the network depth and parameter settings on the model's performance. More analysis on how to improve these aspects would be required.
3. The training data used in the study is limited in diversity, which may affect the model's ability to generalize well to new test areas. Incorporating more diverse landslide samples and expanding the training set with different landslide types and features could improve the model's performance.
4. The manuscript does not explore the performance of deeper network architectures, which could further improve the model's performance.
5. The literature review is limited and could have been expanded to provide a more comprehensive background on existing research in the field, as well as to contextualize the contribution of this paper. Topics regarding Mask R-CNN is an area that is developing very fast. Several latest papers are closely related to this area. The following references, but not limited to (Fu et al., 2022; Kubo et al., 2022; Liu et al., 2022), can be well analyzed to strengthen your research.
[1]. Fu, R., He, J., Liu, G., Li, W. L., Mao, J. Q., He, M. H., & Lin, Y. Y. (2022). Fast Seismic Landslide Detection Based on Improved Mask R-CNN. Remote Sensing, 14(16), 3928. https://doi.org/10.3390/rs14163928
[2]. Kubo, S., Yamane, T., & Chun, P. J. (2022). Study on Accuracy Improvement of Slope Failure Region Detection Using Mask R-CNN with Augmentation Method. Sensors, 22(17), Article 6412. https://doi.org/10.3390/s22176412
[3]. Liu, Y., Yao, X., Gu, Z. K., Zhou, Z. K., Liu, X. H., Chen, X. M., & Wei, S. F. (2022). Study of the Automatic Recognition of Landslides by Using InSAR Images and the Improved Mask R-CNN Model in the Eastern Tibet Plateau. Remote Sensing, 14(14), 3362. https://doi.org/10.3390/rs14143362
SPECIFIC COMMENTS
6. The original conclusion part is too loose and difficult to understand. Note that if an article has too many conclusions, it means that there is no conclusion. At least not with important conclusions. It could be better organized and more concise. It is difficult to follow due to its loose structure and the inclusion of various points within a single paragraph, especially for Lines 509-520. Ideally, it should have a more focused presentation of the main findings, contributions, limitations, and future work. Improving the clarity and structure of the conclusion section would greatly enhance the overall readability and impact of the paper. Please succinctly re-structure the conclusions.
Author Response
Dear reviewer:
Your comments are all valuable and very helpful for revising and improving our paper, as well as the important guiding significance to our research. The attachment file has the responses to your comments on manuscript ID sensors-2270903. We have carefully studied the comments and provided detailed responses and major corrections to each comment for your review.

Reviewer 2 Report
The manuscript is written on a relevant topic. However, before the article is published, minor edits are needed.
Abstract. The abstract should be a total of about 200 words maximum, now its 220.
1. Introduction
The authors provided a fairly complete overview of existing methods on the topic under study.
2. Materials and Methods
Line 156 – paragraph indent
Line 156 – Missed space before formulas.
3. Experiments
Figure 6 - Needs to be redone. The Figure does not reflect the exact location of the survey site in Sanming. It is better to indicate the position of the area under study with a dot at the bottom right figure and draw an arrow to the UAV image. Given that the UAV image is rotated relative to the north, the coordinates must be specified on all sides of the image. The direction to the north does not coincide with the north in Figure 7. The scale bar also differs in size from Fig 7 by 2 times, although the scale of the images is the same. Also, since the image is rotated almost 90 degrees, latitude and longitude are most likely confused. It is necessary to indicate the main settlements on all maps.
Table 1 - The Remark column is not informative and should be removed. Band names can be placed in brackets after their number in the Value column, or you can rename the name of the row to Bands.
Line 276 – What types of landslides, according to Varnes' classification, occur in the study area? Describe the nature of landslide processes, their type, refer to sources or one of them:
Varnes, D.J., 1978, Slope movement types and processes, in Schuster, R.L., and Krizek, R.J., eds., Landslides—Analysis and control: National Research Council, Washington, D.C., Transportation Research Board, Special Report 176, p. 11–33
Cruden D.M., VARNES D. J. (1996) - Landslide types and processes. In: Turner A.K.; Shuster R.L. (eds) Landslides: Investigation and Mitigation. Transp Res Board, Spec Rep 247, pp 36–75.
Line 283-285 - The sentence should be restructured to make it easier to read, or split into two sentences.
Line 285-286 – Why 10 gradations? How is it used in the analysis of the results? There is no standard of landslide size classification and is often considered as terrain specific or at the max while comparing one terrain with that of another. Despite this, there are examples of classifications, maybe you should do a similar one or use existing gradations. For example:
Line 289 - The upper limit of the landslide area range does not match the maximum area value at Line 281.
Figure 7 - Rename legend name - Landslide area, m2. Why is the range 2869.73-4297.22 different from the general color scale? Specify the number of landslides in each range.
Line 305 - Specify the total number of landslide samples, size and percentage of traning set and validation set.
Formula (4) - white spaces in the curly brace. Extra spaces in the words Precision, Recall
Formula (5) - Extra spaces in words Precision, Recall
4. Results Line 405 - All Figures, Schemes and Tables should be inserted into the main text close to their first citation and must be numbered following their number of appearance.. The first reference is on Table 3 and then on Table 2. Change the Table numbers and put them in the text after their citation. Line 425 - Provide a table showing how the improved model recognizes landslides of different size (small, medium, large).
Author Response

(The authors gave the same response as above.)

Round 2
Reviewer 1 Report
The current version has seen significant improvements. In this round, the authors have addressed all five general and one specific comment. They have accepted most of the comments, provided detailed explanations, and incorporated the changes in the manuscript. I have observed a considerable difference between the current version and the original submission, and the paper's quality has steadily improved, reaching the average academic publication level for the journal Sensors. In light of these improvements, I recommend accepting this manuscript in its current form.